# Adversarial bandit optimization for approximately linear functions

## Abstract

We consider a variant of the standard Bandit linear optimization, where in each trial the loss function is the sum of a linear function and a small but arbitrary perturbation chosen after observing the player's choice. We give both expected and high probability regret bounds for the problem. Our result also implies an improved high-probability regret bound for the Bandit linear optimization, a special case with no perturbation. We also give a lower bound on the expected regret.

## 1 Introduction

Bandit optimization is a sequential game between a player and an adversary. The game is played over $T$ rounds, where $T$ is a positive natural number called the horizon. The game is specified by a pair $(\mathcal{K}, \mathcal{F})$, where $\mathcal{K} \subseteq \mathbb{R}^d$ is a bounded closed convex set and $\mathcal{F} \subseteq \{f : \mathcal{K} \to \mathbb{R}\}$ is a function class. In each round $t \in [T]$, the player first chooses an action $x_t \in \mathcal{K}$ and the adversary chooses a loss function $f_t \in \mathcal{F}$, and then the player receives the value $f_t(x_t)$ as the loss. Note that $f_t$ itself is unknown to the player. In this paper, we assume the adversary is oblivious, i.e., the loss functions are specified before starting the game [1]. The goal of the player is to minimize the regret

$$\sum_{t=1}^{T} f_t(x_t) - \min_{x \in \mathcal{K}} \sum_{t=1}^{T} f_t(x) \tag{1}$$

in expectation (expected regret) or with high probability (high-probability regret).

For convex loss functions, the bandit optimization has been extensively studied (see, e.g.,Dani et al. (2007); Abernethy et al. (2008); Lee et al. (2020)). $\mathcal{O}(d^{1/3}T^{3/4})$ regret bounds are shown by Flaxman et al. (2005). Lattimore (2020) shows an information-theoretic regret bound $\widetilde{O}(d^{2.5}\sqrt{T})$ for convex loss functions. For linear loss functions, Abernethy et al. (2008) propose the SCIRBLE algorithm and give an expected regret bound $\mathcal{O}(d\sqrt{T \ln T})$, achieving optimal dependence on $T$(Bubeck et al., 2012). Lee et al. (2020) propose the SCRIBLE with lifting and show a high-probability regret bound $\widetilde{O}(d^2\sqrt{T})$.

Recently, non-convex functions are also getting popular in this literature. For example, Agarwal et al. (2019) show a regret bound $\mathcal{O}(poly(d)T^{2/3})$ for smooth and bounded non-convex functions. Ghai et al. (2022) propose algorithms with regret bounds $\mathcal{O}(poly(d)T^{2/3})$ under the assumption that non-convex functions are reparametlized as some convex functions.

In this paper, we investigate the bandit optimization problem for a class of non-convex non-smooth loss functions. The function class consists of non-smooth and non-convex functions that are "close" to linear functions, in the sense that functions in the class can be viewed as linear functions with adversarial non-convex perturbations whose amount is up to $\epsilon$. Bandit optimization for linear loss functions with stochastic noise (e.g., Abbasi-Yadkori et al. (2011); Amani et al. (2019)) cannot be applied to our problem. Also, standard Bandit linear optimization methods for estimating the gradient, such as self-concordant barrier

---

[1]We do not consider the case where the adversary is adaptive, i.e., it can choose the $t$-th loss function $f_t$ depending on the previous actions $x_1, \ldots, x_{t-1}$.

regularizer(Hazan et al., 2016), cannot be effectively applied to our problem. We propose a novel approach to analyzing high-probability regret, introducing a new method for decomposing regret. Additionally, we propose a novel method to bound $\|x_t - u\|$, where $u \in \mathcal{K}$, to account for the impact of perturbations.

*(margin note: adding a high-level overview by Reviewer iDsE)*

1. When $\epsilon \neq 0$, we propose a modification of the SCRIBLE with lifting and increasing learning rates (Lee et al., 2020) and utilize the properties of the $\nu$-normal barrier(Nemirovski, 2004) to prove its high probability regret bound $\widetilde{O}(d\sqrt{T} + \epsilon dT)$, and we also obtain its expected regret $\mathcal{O}(d\sqrt{T \ln T} + \epsilon dT)$.

*(margin note: Focus on d and T (Literature Review by Reviewer TGdS))*

2. When $\epsilon = 0$, this problem becomes Bandit linear optimization, a special case with no perturbation. Compared to Lee et al. (2020)'s results, holding with probability $1 - \gamma$, $\mathcal{O}(\ln^2(dT)d^2 \ln T \sqrt{T \ln \frac{\ln(dT)}{\gamma}})$, we use a different regret decomposition approach to achieve a better high-probability regret bound $\mathcal{O}(d\sqrt{T \ln T} + \ln T \sqrt{T \ln(\frac{\ln T}{\gamma})} + \ln(\frac{\ln T}{\gamma}))$.

3. We prove a lower bound $\Omega(\epsilon T)$, implying that our bounds are tight w.r.t. the parameter $\epsilon$.

## 2 Related Work

The bandit linear optimization was first proposed by Awerbuch & Kleinberg (2004), who achieved a regret bound of $\mathcal{O}(d^{3/5}T^{2/3})$ against an oblivious adversary. Later, McMahan & Blum (2004) established a regret bound of $\mathcal{O}(dT^{3/4})$ when facing an adaptive adversary. A foundational approach in bandit optimization problems involves gradient-based smoothing techniques. Abernethy et al. (2012) presented pioneering work in this area and achieved an expected regret bound of $\mathcal{O}(d\sqrt{T \ln T})$ when dealing with an oblivious adversary. Bartlett et al. (2008) proposed a high-probability regret bound of $\mathcal{O}(d^{2/3}\sqrt{T \ln dT})$ under a special condition. Subsequently, Lee et al. (2020) presented a high-probability regret bound $\widetilde{O}(d^2\sqrt{T})$ for both oblivious and adaptive adversaries.

*(margin note: Add some references (B:missing... by Reviewer iDsE))*

Unlike convex bandit problems, which have been extensively explored and analyzed, non-convex bandits introduce unique challenges due to the complexity of exploring and exploiting in a non-convex area. Gao et al. (2018) considered both non-convex losses and non-stationary data and established a regret bound of $O(\sqrt{T + poly(T)})$. Yang et al. (2018) achieved a regret bound of $O(\sqrt{T \log T})$ for a non-convex loss functions. However, they both required the loss functions to have smoothness properties, and our loss functions are neither convex nor smooth.

### 2.1 Comparison to Lee et al. (2020)

Our approach builds upon Lee et al. (2020)'s work. Below, we highlight the key differences between our method and Lee et al. (2020)'s in the context of the oblivious bandit setting:

*(margin note: Add comparison (Weaknesse1 by Reviewer 6Zex and Literature Review by Reviewer TGdS))*

1. Simplified Regret Analysis: While Lee's regret analysis introduces unnecessary complexity for the oblivious bandit setting, our approach simplifies the analysis, leading to more streamlined results.

2. Reduced Dependence on $d$: Lee et al. (2020)'s analysis results in a regret bound with greater dependence on $d$, whereas our method derives a bound with significantly reduced dependence on $d$ (This distinction is demonstrated in the introduction and further illustrated in the case where $\epsilon = 0$).

3. Revised Generality of Problem Setting: Like the SCIRBLE algorithm, our approach is more general, treating bandit linear optimization as a special case within a broader problem framework.

## 3 Preliminaries

This section introduces some necessary notations and defines $\epsilon$-approximately linear function. Then we give our problem setting.

## 3.1 Notation

We abbreviate the 2-norm $\|\cdot\|_2$ as $\|\cdot\|$. For a twice differentiable convex function $\mathcal{R} : \mathbb{R}^d \to \mathbb{R}$ and any $x, h \in \mathbb{R}^d$, let $\|h\|_x = \|h\|_{\nabla^2 \mathcal{R}(x)} = \sqrt{h^\top \nabla^2 \mathcal{R}(x) h}$, and $\|h\|_x^* = \|h\|_{(\nabla^2 \mathcal{R}(x))^{-1}} = \sqrt{h^\top (\nabla^2 \mathcal{R}(x))^{-1} h}$, respectively.

For any $v \in \mathbb{R}^d$, let $v^\perp$ be the space orthogonal to $v$. Let $\mathbb{S}_1^d = \{x \mid \|x\| = 1\}$. The vector $e_i \in \mathbb{R}^d$ is a standard basis vector with a value of 1 in the $i$-th position and 0 in all other positions. $I$ is an identity matrix with dimensionality implied by context.

## 3.2 Problem Setting

Let $\mathcal{K} \subseteq \mathbb{R}^d$ be a bounded and closed convex set such that for any $x, y \in \mathcal{K}$, $\|x - y\| \leq D$. Furthermore, we assume that $\mathcal{K}$ contains the unit ball centered at the zero vector. Otherwise, we can apply an affine transformation to translate the center point of the convex set to the origin. Let $\mathcal{K}' = \{(x, 1) : x \in \mathcal{K}\}$. For any $\delta \in (0, 1)$, let $\mathcal{K}_\delta = \{x | \frac{1}{1-\delta} x \in \mathcal{K}\}$ and $\mathcal{K}_\delta' = \{(x, 1) : x \in \mathcal{K}_\delta\}$, respectively.

**Definition 1.** *A function $f : \mathcal{K} \to \mathbb{R}$ is $\epsilon$-approximately linear if there exists $\theta_f \in \mathbb{R}^d$ such that $\forall x \in \mathcal{K}$, $|f(x) - \theta_f^\top x| \leq \epsilon$.*

For convenience, in the definition above, let $\sigma_f(x) = f(x) - \theta_f^\top x$, and we omit the subscript $f$ of $\theta_f$ and $\sigma_f$ if the context is clear. Note that $|\sigma(x)| \leq \epsilon$ for any $x \in \mathcal{K}$.

In this paper, we consider the bandit optimization $(\mathcal{K}, \mathcal{F})$, where $\mathcal{F}$ is the set of $\epsilon$-approximately linear functions $f(x) = \theta^\top x + \sigma(x)$ with $\|\theta\| \leq G$.

The bandit optimization for $\epsilon$-approximately linear functions can be defined as the following statement. For every round $t = 1, .., T$, the player chooses an action $x_t \in \mathcal{K}$, and the adversary simultaneously chooses a linear function $f_t(x_t)(= \theta_t^\top x_t)$ at the same time. After observing the player's choice $x_t$, the adversary chooses a perturbation $\sigma_t(x_t)(|\sigma_t(x_t)| \leq \epsilon)$. The value of $\epsilon$-approximately linear function $f_t(x_t) = \theta_t^\top x_t + \sigma_t(x_t)$ is then revealed to the player. The goal of the player is to minimize the regret $\sum_{t=1}^T f_t(x_t) - \min_{x \in \mathcal{K}} \sum_{t=1}^T f_t(x)$.

*Define sigma_t and theta_t ( Writing Quality3 by Reviewer iDsE)*

# 4 Main Results

*Add more descriptions(Clarity in Methods by Reviewer TGds*

In this section, we first review SCRIBLE and SCRIBLE with lifting and increasing learning rates, followed by presenting the main contributions of this paper with detailed explanations.

## 4.1 SCRIBLE

*Reoganize section 4(Unorganized Lemmas and Theorems by Reviewer TGdS)*

When $\epsilon = 0$, our problem becomes bandit linear optimization problem, which is a problem that has been widely studied. To introduce our result, we start by revisiting the SCRIBLE algorithm, which is effective for the bandit linear optimization problem. It utilizes a $\kappa$-self concordant barrier $\mathcal{T}$ (Abernethy et al., 2008), which always exists on $\mathcal{K}$. At each round $t$, the player chooses a point $x_t \in \mathcal{K}$, but the actual point played is $y_t = x_t + \mathbf{A}_t \mu_t$, where $\mathbf{A}_t = [\nabla^2 \mathcal{T}(x_t)]^{-\frac{1}{2}}$ and $\mu_t$ is uniformly randomly sampled from the $d$-dimensional unit sphere $\mathbf{S}^d$. It maintains a sequence $x_1, ..., x_T \in \mathcal{K}$, which is updated according to the rule: $x_{t+1} = \arg\min_{x \in \mathcal{K}} \eta \sum_{\tau=1}^t g_\tau^\top x + \mathcal{T}(x)$, where $g_\tau$ is an estimator for $\theta_\tau$, $\eta$ is some learning rate. It relies only on the feedback $f_t(y_t)(= \theta_t^\top y_t)$ to build an unbiased estimator $g_t = df_t(y_t)\mathbf{A}_t^{-1}\mu_t$ for $\theta_t$.

*Add description of SCRIBLE and SCRIBLE with ..rates( Writing Quality1 by Reviewer iDsE)*

In the analysis (Abernethy et al., 2008) of SCRIBLE, the unbiased estimator $g_t$ helps establish that for any $u \in \mathcal{K}$, $\theta_t^\top(x_t - u) = \mathbb{E}_t[g_t^\top(x_t - u)] \leq \mathcal{O}(\eta d^2)$. This result is instrumental in deriving the regret bound $\mathcal{O}(\frac{\kappa}{\eta} \log T + \eta d^2 T)$ for oblivious adversary. However, in our problem setting, the estimates $g_t = d[\theta_t^\top y_t + \sigma_t(y_t)]\mathbf{A}_t^{-1}\mu_t$ obtained using SCRIBLE will always be influenced by $\sigma_t(y_t)$. This necessitates additional focus on bounding $(d\sigma_t(y_t)\mathbf{A}_t^{-1}\mu_t)^\top(x_t - u)$. Although the Cauchy-Schwarz inequality helps in deriving the bounds $(d\sigma_t(y_t)\mathbf{A}_t^{-1}\mu_t)^\top(x_t - u) \leq \|d\sigma_t(y_t)\mathbf{A}_t^{-1}\mu_t\|_{\nabla^2 \mathcal{T}(x_t)}^* \|x_t - u\|_{\nabla^2 \mathcal{T}(x_t)}$ and $\|d\sigma_t(y_t)\mathbf{A}_t^{-1}\mu_t\|_{\nabla^2 \mathcal{T}(x_t)}^* \leq d\epsilon$ (a similar proof can be found in Equations 24 to 30), the analysis of SCRIBLE, particularly the $\kappa$-self concordant barrier $\mathcal{T}$, cannot help us bound $\|x_t - u\|_{\nabla^2 \mathcal{T}(x_t)}$. Furthermore, since the largest eigenvalue

of $\nabla^2 \mathcal{T}(x_t)$(Nemirovski, 2004) could potentially approach infinity, bounding $\|x_t - u\|_{\nabla^2 \mathcal{T}(x_t)}$ becomes a significant challenge.

## 4.2 SCRIBLE with lifting and increasing learning rates

SCRIBLE with lifting and increasing learning rates introduces a dummy coordinate with a value of 1 appended to all actions, resulting in the lifted decision set $\mathcal{K}' = \{(x,1) : x \in \mathcal{K}\}$. This transformation lifts the bandit linear optimization problem to $\mathbb{R}^{d+1}$. The conic hull of this set is $con(\mathcal{K}) = \{\mathbf{0}\} \cup \{(x,b) : \frac{x}{b} \in \mathcal{K}, x \in \mathbb{R}^d, b > 0\}$.

The algorithm performs SCRIBLE over the lifted decision set, using a $\nu$-normal barrier $\mathcal{R}$ defined over the $con(\mathcal{K})$ (which always exists) as the regularizer to generate the sequence $x'_1, ..., x'_t$. It set $y'_t = x'_t + \mathbf{A}_t \mu_t$ where $\mathbf{A}_t = [\nabla^2 \mathcal{R}(x'_t)]^{-\frac{1}{2}}$ and $\mu_t$ is uniformly sampled at random from $\mathbb{S}_1^{d+1} \cap (\mathbf{A}_t e_{d+1})^\perp$. Since $\mu_t$ is orthogonal to $\mathbf{A}_t e_{d+1}$, the last coordinate of $\mathbf{A}_t \mu_t$ is zero, ensuring that $y'_t = (y_t, 1)$ remains within $\mathcal{K}'$. The actual point played is still $y_t$. After playing $y_t$ and observing $f_t (= \theta_t^\top y_t)$, it constructs the loss estimator the same way as SCRIBLE: $g_t = df_t(y_t) \mathbf{A}_t^{-1} \mu_t$. The analysis by Lee et al. (2020) shows that the first $d$ coordinates of $g_t$ are indeed an unbiased estimator of $\theta_t$.

SCRIBLE with lifting and increasing learning rates utilizes the properties of the $\nu$-normal barrier $\mathcal{R}$, which are not available in the $\kappa$-self concordant barrier $\mathcal{T}$, as well as increasing learning rates, to bound $\|h\|_{\nabla^2 \mathcal{R}(x'_t)} \leq -\|h\|_{\nabla^2 \mathcal{R}(x'_{t+1})} + \nu \ln(\nu T + 1)$, where $h \in \mathcal{K}'$. Besides, Lee et al. (2020) divide the regret mainly into two parts: $\sum_{t=1}^T [y_t^\top \theta_t - x_t^\top g_t + u^\top (g_t - \theta_t)]$ and $\sum_{t=1}^T (y_t - u)^\top g_t$, where $u \in \mathcal{K}$. They provide a bound for $\sum_{t=1}^T (y_t - u)^\top g_t$ and a high-probability bound for $\sum_{t=1}^T [y_t^\top \theta_t - x_t^\top g_t + u^\top (g_t - \theta_t)]$, which helps derive high-probability regret bounds for both oblivious and adaptive adversaries. However, for our problem, the bound they obtain for $\|h\|_{\nabla^2 \mathcal{R}(x'_t)}$ is still too large and does not help in bounding $\|x'_t - h\|_{\nabla^2 \mathcal{R}(x'_t)}$. Again, since the largest eigenvalue of $\nabla^2 \mathcal{R}(x_{t'})$(Nemirovski, 2004) could potentially approach infinity, bounding $\|x'_t - h\|_{\nabla^2 \mathcal{R}(x'_t)}$ remains a challenge, and no previous work has addressed this.

## 4.3 SCRIBLE with lifting

For a decision set $\mathcal{K}$ with a $\nu$-normal barrier on $con(\mathcal{K})$, where $con(\mathcal{K}) = \{\mathbf{0}\} \cup \{(x,b) : \frac{x}{b} \in \mathcal{K}, x \in \mathbb{R}^d, b > 0\}$, we apply Algorithm 1 to approximately linear functions. Recall $\mathcal{K}' = \{(x,1) : x \in \mathcal{K}\}$.

We simplify the SCRIBLE with lifting and increasing learning rates(Lee et al., 2020). We do not use the increasing learning rates part but retain the lifting. This preserves its advantages; for instance, the $\nu$-normal barrier $\mathcal{R}$ always exists on $con(\mathcal{K})$. Additionally, the actual point played by the algorithm is $y_t$ and $y'_t = (y_t, 1) = x'_t + \mathbf{A}_t \mu_t$ always remains within $\mathcal{K}'$. Furthermore, it adopts the same update method as FTRL algorithm(Hazan et al., 2016): $x'_{t+1} = \arg \min_{x' \in \mathcal{K}'} \eta \sum_{\tau=1}^t g_\tau^\top x' + \mathcal{R}(x')$. Although it constructs the same loss estimator, $g_t = df_t(y_t) \mathbf{A}_t^{-1} \mu_t$ as the original algorithm, $g_t$ is no longer an unbiased estimator of $\theta_t$

---

**Algorithm 1** SCRIBLE with lifting

**Input:** $T$, parameters $\eta \in \mathbb{R}, \delta \in (0,1)$, $\nu$-normal barrier $\mathcal{R}$ on $con(\mathcal{K})$
1: Initialize: $x'_1 = \arg \min_{x' \in \mathcal{K}'} \mathcal{R}(x')$
2: **for** $t = 1, .., T$ **do**
3:      let $\mathbf{A}_t = [\nabla^2 \mathcal{R}(x'_t)]^{-\frac{1}{2}}$
4:      Draw $\mu_t$ from $\mathbb{S}_1^{d+1} \cap (\mathbf{A}_t e_{d+1})^\perp$ uniformly, set $y'_t = (y_t, 1) = x'_t + \mathbf{A}_t \mu_t$.
5:      Play $y_t$, observe and incur loss $f_t(y_t)$. Let $g_t = df_t(y_t) \mathbf{A}_t^{-1} \mu_t$.
6:      Update $x'_{t+1} = \arg \min_{x' \in \mathcal{K}'} \eta \sum_{\tau=1}^t g_\tau^\top x' + \mathcal{R}(x')$
7: **end for**

---

We present our main results: expected and high-probability regret bounds for the problem.

Prensent Theo1 and Theo2 firstly ( Writing Quality2 by Reviewer iDsE)

**Theorem 1.** *The algorithm with parameters $\eta = \frac{\sqrt{2\nu \log T}}{2d\sqrt{T}}, \delta = \frac{1}{T^2}$ guarantees the following expected regret bound*

$$\mathbb{E}[\sum_{t=1}^{T} f_t(y_t) - \min_{x \in \mathcal{K}} \sum_{t=1}^{T} f_t(x)] \le 4d\sqrt{2\nu T \log T} + \frac{GD}{T} + 2T\epsilon + dT\epsilon(2\nu + \sqrt{\nu}). \tag{2}$$

**Theorem 2.** *The algorithm with parameters $\eta = \frac{\sqrt{2\nu \ln T}}{2d\sqrt{T}}, \delta = \frac{1}{T^2}$ ensures that with probability at least $1 - \gamma$*

$$\sum_{t=1}^{T} f_t(y_t) - \min_{x \in \mathcal{K}} \sum_{t=1}^{T} f_t(x) \le 4d\sqrt{2\nu T \ln T} + \frac{GD}{T} + Td\epsilon(2\nu + \sqrt{\nu}) + C(1+\epsilon)\sqrt{8T \ln \frac{C}{\gamma}} + 2GD \ln \frac{C}{\gamma} + 2T\epsilon \tag{3}$$

*where $C = \lceil \ln GD \rceil \lceil \ln((GD)^2 T) \rceil$.*

We primarily divide the regret into parts $\sum_{t=1}^{T}(x_t - u)^\top \theta_t$ and $\sum_{t=1}^{T}(y_t - x_t)^\top \theta_t$ rather than following the approach of Lee et al. (2020), where $u \in \mathcal{K}$. This means that, for an oblivious adversary, calculating the regret does not require considering the variance of the estimator $g_t$ and $\theta_t$, but only the variance between $y_t$ and $x_t$. This difference is a key factor that enables us to achieve a better high-probability regret bound when $\epsilon = 0$.

Firstly, we bound $\sum_{t=1}^{T}(x_t - u)^\top \theta_t$. As mentioned earlier, for $\epsilon$-approximately linear functions, bounding $\sum_{t=1}^{T}(x_t - u)^\top \theta_t$ requires considering the term $(d\sigma_t(y_t)\mathbf{A}_t^{-1}\mu_t)^\top (x_t - u)$. The main difficulty in bounding $(d\sigma_t(y_t)\mathbf{A}_t^{-1}\mu_t)^\top (x_t - u)$ is that the norm $\|x_t - u\|$ is hard to bound. Thus, we make the entire analysis hold in $\mathbb{R}^{d+1}$ and transform the problem from bounding the norm $\|x_t - u\|$ to how to bound the norm $\|x'_t - h\|$, where $h \in \mathcal{K}'$. We present a straightforward yet necessary Lemma 4, which helps to bound $\|h\|_{\nabla^2 \mathcal{R}(x'_t)} \le 2\nu$. In addition, the properties of the $\nu$ normal barrier tell us $\|x'_t\|_{\nabla^2 \mathcal{R}(x'_t)} = \sqrt{\nu}$. With these two conditions, we can immediately deduce the bound $\|x'_t - h\|_{\nabla^2 \mathcal{R}(x'_t)}$ as $2\nu + \sqrt{\nu}$. This also implies that increasing learning rates are not required in our case, as they are solely aimed at controlling $\|h\|_{\nabla^2 \mathcal{R}(x'_t)} \le -\|h\|_{\nabla^2 \mathcal{R}(x'_{t+1})} + \nu \ln(\nu T + 1)$ in Lee et al. (2020)'s paper.

Secondly, by obtaining the expected bound and high-probability bound for $\sum_{t=1}^{T}(y_t - x_t)^\top \theta_t$, we can derive the expected regret bound and high-probability regret bound, respectively. For high-probability bound of $\sum_{t=1}^{T}(y_t - x_t)^\top \theta_t$, unlike SCRIBLE with lifting and increasing learning rates, which constrains the decision set from $\mathcal{K}'$ to $\mathcal{K}'_\delta$ to ensure that $x'_t$ is never too close to the boundary (thus ensuring that the eigenvalues of $\mathbf{A}_t$ are bounded, especially for bounding $\|h\|_{\nabla^2 \mathcal{R}(x'_t)}$). Our approach does not require $x'_t$ to stay away from the boundary. Furthermore, we do not need to bound the eigenvalues of $\mathbf{A}_t$, which gives us greater flexibility in choosing the value of $\delta$ (such as $\frac{1}{T^2}$), leading to a better upper bound for the regret.

Finally, we prove the lower bound of regret in section 6.

# 5 Proof

This section introduces the preliminary of $\nu$-normal barrier, presents several essential lemmas, and gives the proof of main theorems.

## 5.1 $\nu$-normal barrier

We introduce the $\nu$-normal barrier, providing its definitions and highlighting several key properties that will be frequently used in the subsequent analysis.

**Definition 2.** *Let $\Psi \in \mathbb{R}^d$ be a closed and proper convex cone and let $\nu \ge 1$. A function $\mathcal{R} : int(\Psi) \to \mathbb{R}$: is called a $\nu$-logarithmically homogeneous self-concordant barrier (or simply $\nu$-normal barrier) on $\Psi$ if*

    *1. $\mathcal{R}$ is three times continuously differentiable and convex and approaches infinity along any sequence of points approaching the boundary of $\Psi$.*

*(margin note: Add explation (C:Some... by Reviewer iDsE and Clarity in Methods by Reviewer TGdS))*

*(margin note: Add more description(in new section 5) bewteen threeoms(Weak nesses2 by Reviewer 6Zex and Unorganized ..Theorems by Reviewer TGdS))*

*(margin note: Add a subsection about normal barrier(Unorg anized ..Theorems by Reviewer TGdS))*

2. *For every $h \in \mathbb{R}^d$ and $x \in int(\Psi)$ the following holds:*

$$\sum_{i=1}^{d}\sum_{j=1}^{d}\sum_{k=1}^{d} \frac{\partial^3 \mathcal{R}(x)}{\partial x_i \partial x_j \partial x_k} h_i h_j h_k \leq 2\|h\|_x^3, \tag{4}$$

$$|\nabla \mathcal{R}(x)^\top h| \leq \sqrt{\nu}\|h\|_x, \tag{5}$$

$$\mathcal{R}(tx) = \mathcal{R}(x) - \nu \ln t, \forall x \in int(\Psi), t > 0. \tag{6}$$

**Lemma 1** (Nemirovski (2004); Nesterov & Nemirovskii (1994)). *If $\mathcal{R}$ is a $\nu$-normal barrier on $\Psi$, Then for any $x \in int(\Psi)$ and any $h \in \Psi$, we have*

$$\|x\|_x^2 = \nu, \tag{7}$$

$$\nabla^2 \mathcal{R}(x)x = -\nabla \mathcal{R}(x), \tag{8}$$

$$\|h\|_x \leq -\nabla \mathcal{R}(x)^\top h, \tag{9}$$

$$\nabla \mathcal{R}(x)^\top (h - x) \leq \nu. \tag{10}$$

**Lemma 2** (Nemirovski (2004)). *If $\mathcal{R}$ is a $\nu$-normal barrier on $\Psi$, then the Dikin ellipsoid centered at $x \in int(\Psi)$, defined as $\{y : \|y - x\|_x \leq 1\}$, is always within $\Psi$. Moreover,*

$$\|h\|_y \geq \|h\|_x (1 - \|y - x\|_x) \tag{11}$$

*holds for any $h \in \mathbb{R}^d$ and any $y$ with $\|y - x\|_x \leq 1$.*

**Lemma 3** (Hazan et al. (2016)). *Let $\mathcal{R}$ is a $\nu$-normal barrier over $\Psi$, then for all $x, z \in int(\Psi) : \mathcal{R}(z) - \mathcal{R}(x) \leq \nu \log \frac{1}{1 - \pi_x(z)}$, where $\pi_x(z) = \inf\{t \geq 0 : x + t^{-1}(z - x) \in \Psi\}$.*

$\nu$-normal barrier plays a crucial role in addressing one of the key challenges in this problem: bounding $\|x_t' - h\|_{x_t'}$, where $h \in \mathcal{K}'$. Equation 7 give $\|x_t'\|_{x_t'} = \sqrt{\nu}$. Building upon Lemma 1, we introduce an effective Lemma 4 that aids in bounding $\|h\|_{x_t'}$.

**Lemma 4.** *If $\mathcal{R}$ be a $\nu$-normal barrier for $\Psi \subseteq R^d$, then for any $x \in int(\Psi)$ and any $h \in \Psi$, we have*

$$\|h\|_x \leq 2\nu. \tag{12}$$

*Proof.* From Lemma 1, we have

$$\|h\|_x \leq -\nabla \mathcal{R}(x)^\top h \leq |\nabla \mathcal{R}(x)^\top h|. \tag{13}$$

Then,

$$|\nabla \mathcal{R}(x)^\top h| = |\nabla \mathcal{R}(x)^\top (h - x + x)| \leq |\nabla \mathcal{R}(x)^\top (h - x)| + |\nabla \mathcal{R}(x)^\top x|. \tag{14}$$

By Lemma 1, $|\nabla \mathcal{R}(x)^\top (h - x)| + |\nabla \mathcal{R}(x)^\top x| \leq \nu + |x^\top \nabla^2 \mathcal{R}(x)x| = 2\nu$. □

With the help of Equation 7 and Lemma 4, it is easy to apply the triangle inequality to derive $\|x_t' - h\|_{x_t'} \leq \|x_t'\|_{x_t'} + \|h\|_{x_t'} \leq \sqrt{\nu} + 2\nu$, where $h \in \mathcal{K}'$.

### 5.2 Useful lemmas

In addition to the properties of the normal barrier and its related lemmas, we also need to introduce some additional necessary lemmas.

Like Lemma 6 in the SCRIBLE algorithm(Abernethy et al., 2008), the next minimizer $x_{t+1}'$ is "close" to $x_t'$. However, there are two differences here: the first is that $\nabla \phi_{t-1}(x_t') \neq 0$ is possible, where $\phi_t(x') = \eta \sum_{\tau=1}^{t} g_\tau^\top x' + \mathcal{R}(x')$. And the second is that for $z = x_t' + \alpha u$, where $u$ is a vector such that $\|u\|_{x_t'} = 1$ and $\alpha \in (-\frac{1}{2}, \frac{1}{2})$, we need to satisfy $z \in \mathcal{K}'$ instead of $z \in \mathcal{K}$.

**Lemma 5.** $x_{t+1}' \in W_{4d\eta}(x_t')$, *where* $W_r(x') = \{y \in \mathcal{K}' : \|y - x'\|_{x'} < r\}$.

*Proof.* Recall that $x'_{t+1} = \arg\min_{x' \in \mathcal{K}'} \phi_t(x')$, where $\phi_t(x') = \eta \sum_{\tau=1}^t g_\tau^\top x' + \mathcal{R}(x')$. Let $h_t(x) = \phi_t((x,1)) = \phi_t(x')$, then $\min h_t(x) = \min \phi_t(x')$. Noticing that $h_t$ is a convex function on $\mathbb{R}^d$ and still holds the barrier property(approaches infinity along any sequence of points approaching the boundary of $\mathcal{K}$). By properties of convex functions, we can get $\nabla h_{t-1}(x_t) = 0$ and for the first $d$ coordinates $\nabla \phi_{t-1}(x'_t) = 0$.

Consider any point in $z \in W_{\frac{1}{2}}(x'_t)$. It can be written as $z = x'_t + \alpha u$ for some vector $u$ such that $\|u\|_{x'_t} = 1$ and $\alpha \in (-\frac{1}{2}, \frac{1}{2})$. Noticing the $d+1$ coordinate of $u$ is 0. Expanding,

$$
\begin{aligned}
\phi_t(z) &= \phi_t(x'_t + \alpha u) \\
&= \phi_t(x'_t) + \alpha \nabla \phi_t(x'_t)^\top u + \alpha^2 \frac{1}{2} u^\top \nabla^2 \phi_t(\xi) u \\
&= \phi_t(x'_t) + \alpha (\nabla \phi_{t-1}(x'_t) + \eta g_t)^\top u + \alpha^2 \frac{1}{2} u^\top \nabla^2 \phi_t(\xi) u \\
&= \phi_t(x'_t) + \alpha \eta g_t^\top u + \alpha^2 \frac{1}{2} u^\top \nabla^2 \phi_t(\xi) u,
\end{aligned}
$$

for some $\xi$ on the path between $x'_t$ and $x'_t + \alpha u$ and the last equality holds because $\nabla \phi_{t-1}(x'_t)^\top u = 0$. Setting the derivative with respect to $\alpha$ to zero, we obtain

$$
|\alpha^*| = \frac{\eta |g_t^\top u|}{u^\top \nabla^2 \phi_t(\xi) u} = |\alpha^*| = \frac{\eta |g_t^\top u|}{u^\top \nabla^2 \mathcal{R}(\xi) u} \tag{15}
$$

The fact that $\xi$ is on the line $x'_t$ to $x'_t + \alpha u$ implies that $\|\xi - x'_t\|_{x'_t} \le \|\alpha u\|_{x'_t} \le \frac{1}{2}$. Hence, by Lemma 2

$$
\nabla^2 \mathcal{R}(\xi) \succeq (1 - \|\xi - x'_t\|_{x'_t})^2 \nabla^2 \mathcal{R}(x'_t) \succ \frac{1}{4} \nabla^2 \mathcal{R}(x'_t). \tag{16}
$$

Thus $u^\top \nabla^2 \mathcal{R}(\xi) u > \frac{1}{4} \|u\|_{x'_t} = \frac{1}{4}$, and $\alpha^* < 4\eta |g_t^\top u|$. Using assumption $\max_{x \in \mathcal{K}} |f_t(x)| \le 1$,

$$
g_t^\top u \le \|g_t\|_{x'_t}^* \|u\|_{x'_t} \le \|d f_t(y_t) \mathbf{A}_t^{-1} \mu_t\|_{x'_t}^* \le \sqrt{d^2 \mu_t^\top \mathbf{A}_t^{-\top} (\nabla^2 \mathcal{R}(x'_t))^{-1} \mathbf{A}_t^{-1} \mu_t} \le d, \tag{17}
$$

we conclude that $|g_t^\top u| \le d$, and $|\alpha^*| < 4d\eta < \frac{1}{2}$ by our choice of $\eta$ and $T$. We conclude that the local optimum $\arg\min z \in W_{\frac{1}{2}(x'_t)} \phi_t(z)$ is strictly inside $W_{4d\eta}(x'_t)$, and since $\phi_t$ is convex, the global optimum is

$$
x_{t+1} = \arg\min_{z \in \mathcal{K}'} \phi_t(z) \in W_{4d\eta}(x'_t). \tag{18}
$$

$\square$

Lemma 5 implies $\|x'_{t+1} - x'_t\|_{x'_t} \le 4d\eta$. This result will help us bound $g_t^\top(x'_t - h)$, where $h \in \mathcal{K}'$(see Lemma 7).

This next lemma is based on Lemma B.9.(Lee et al., 2020), but due to the differences in the loss functions, what we obtain is an unbiased estimate regarding $g_{t,i}$ rather than $\theta_{t,i}$, for $i \in [d]$. Lee et al. (2020) state that $\mathbb{E}_t[l_{t,i}] = \theta_{t,i}$, for $i \in [d]$. Since $l_t = d(\theta_t, 0)(x'_t + \mathbf{A}_t \mu_t) \mathbf{A}_t^{-1} \mu_t$ is identical to ours, we directly apply it to our analyze.

**Lemma 6.** *Let $l_t = d(\theta_t, 0)(x'_t + \mathbf{A}_t \mu_t) \mathbf{A}_t^{-1} \mu_t$. For Algorithm 1, we have $\mathbb{E}_t[l_{t,i}] = \theta_{t,i}$, for $i \in [d]$.*

The regret bound of FTRL algorithm(Hazan et al., 2016) states that for every $u \in \mathcal{K}$, $\sum_{t=1}^T \nabla_t^\top x_t - \sum_{t=1}^T \nabla_t^\top u \le \sum_{t=1}^T [\nabla_t^\top x_t - \nabla_t^\top x_{t+1}] + \frac{1}{\eta}[\mathcal{R}(u) - \mathcal{R}(x_1)]$, where $\nabla_t$ represents the gradient of the loss function $f_t$. In our adaptation, we replaced $\nabla_t$ with $g_t$ and $\mathcal{K}$ with $\mathcal{K}'$. This modification does not fundamentally alter the original result. Since the update way $x'_{t+1} = \arg\min_{x' \in \mathcal{K}'} \eta \sum_{\tau=1}^t g_\tau^\top x' + \mathcal{R}(x')$ satisfied the condition of FTRL algorithm(Hazan et al., 2016), we can apply (Lemma 5.3. in Hazan et al. (2016)) to Algorithm 1 as follow.

**Lemma 7.** *For Algorithm 1 and for every $h \in \mathcal{K}'$, $\sum_{t=1}^T g_t^\top x'_t - \sum_{t=1}^T g_t^\top h \le \sum_{t=1}^T [g_t^\top x'_t - g_t^\top x'_{t+1}] + \frac{1}{\eta}[\mathcal{R}(h) - \mathcal{R}(x'_1)]$.*

The following lemma represents a key proof of this paper. Specifically, it provides a bound for $\sum_{t=1}^{T} \theta_t^\top x_t - \sum_{t=1}^{T} \theta_t^\top x^*$. Due to Lemma 8, we only need to consider $\sum_{t=1}^{T} \theta_t^\top y_t - \sum_{t=1}^{T} \theta_t^\top x_t$ when calculating the regret bound. This result plays a crucial role in deriving both expected and high-probability regret bounds.

**Lemma 8.** *For Algorithm 1, let $f_t(x_t) = \theta_t^\top x_t + \sigma_t(x_t)$ and $x^* = \arg\min_{x \in \mathcal{K}} \sum_{t=1}^{T} f_t(x)$ and we have*

$$\sum_{t=1}^{T} \theta_t^\top x_t - \sum_{t=1}^{T} \theta_t^\top x^* \leq 2\eta d^2 T + \frac{\nu \log(\frac{1}{\delta})}{\eta} + Td\epsilon(2\nu + \sqrt{\nu}) + \delta DGT. \tag{19}$$

*Proof.* Recall for any $\delta \in (0,1)$, $\mathcal{K}_\delta = \{x | \frac{1}{1-\delta} x \in \mathcal{K}\}$ and $\mathcal{K}_\delta' = \{(x,1) : x \in \mathcal{K}_\delta\}$. Let $x_\delta^* = \prod_{\mathcal{K}_\delta} x^*$, by properties of projections, then

$$\|x^* - x_\delta^*\| = \min_{a \in \mathcal{K}_\delta} \|x^* - a\|. \tag{20}$$

Since $(1-\delta)x^* \in \mathcal{K}_\delta$, then

$$\min_{a \in \mathcal{K}} \|x^* - a\| \leq \|x^* - (1-\delta)x^*\| \leq \delta D. \tag{21}$$

So,

$$\|x_\delta^* - x^*\| \leq \delta D. \tag{22}$$

By Cauchy–Schwarz inequality and the fact that $\|\theta\| \leq G$ and $\|x_\delta^* - x^*\| \leq \delta D$,

$$\sum_{t=1}^{T} \theta_t^\top x_\delta^* - \sum_{t=1}^{T} \theta_t^\top x^* \leq \delta DGT. \tag{23}$$

So,

$$\begin{aligned}
\sum_{t=1}^{T} \theta_t^\top x_t - \sum_{t=1}^{T} \theta_t^\top x^* &= \sum_{t=1}^{T} \theta_t^\top x_t - \sum_{t=1}^{T} \theta_t^\top x_\delta^* + \sum_{t=1}^{T} \theta_t^\top x_\delta^* - \sum_{t=1}^{T} \theta_t^\top x^* \\
&\leq \sum_{t=1}^{T} \theta_t^\top x_t - \sum_{t=1}^{T} \theta_t^\top x_\delta^* + \delta DGT.
\end{aligned}$$

Let $\theta_t' = (\theta_t, z)$, where $z$ is the $(d+1)$th coordinate of $d\mathbb{E}_t[(\theta_t, 0)^\top (x_t' + \mathbf{A}_t \mu_t) \mathbf{A}_t^{-1} \mu_t]$. From Lemma 6, we know $d\mathbb{E}_t[(\theta_t, 0)^\top (x_t' + \mathbf{A}_t \mu_t) \mathbf{A}_t^{-1} \mu_t] = \theta_t'$. Since $g_t = df(y_t) \mathbf{A}_t^{-1} \mu_t = d\theta_t'^\top (x_t' + \mathbf{A}_t \mu_t) \mathbf{A}_t^{-1} \mu_t + d\sigma_t(y_t) \mathbf{A}_t^{-1} \mu_t$, and let $M_t = \mathbb{E}_t[d\sigma_t(y_t) \mathbf{A}_t^{-1} \mu_t]$, then $\theta_t' = \mathbb{E}_t[g_t] - M_t$ and that we have

$$\begin{aligned}
\sum_{t=1}^{T} \theta_t^\top x_t - \sum_{t=1}^{T} \theta_t^\top x_\delta^* &= \sum_{t=1}^{T} (\theta_t'^\top x_t' - z) - \sum_{t=1}^{T} (\theta_t'^\top x_\delta^{*'} - z) \\
&= \sum_{t=1}^{T} (\mathbb{E}_t[g_t] - M_t)^\top x_t' - \sum_{t=1}^{T} (\mathbb{E}_t[g_t] - M_t)^\top x_\delta^{*'} \\
&= \sum_{t=1}^{T} \mathbb{E}_t[g_t]^\top (x_t' - x_\delta^{*'}) + \sum_{t=1}^{T} M_t^\top (x_\delta^{*'} - x_t').
\end{aligned}$$

We bound $\sum_{t=1}^{T} M_t^\top (x_\delta^{*'} - x_t')$ firstly. By Cauchy–Schwarz inequality,

$$\sum_{t=1}^{T} M_t^\top (x_\delta^{*'} - x_t') \leq \sum_{t=1}^{T} \|M_t\|_{x_t'}^* \|x_\delta^{*'} - x_t'\|_{x_t'}.$$

By Jensen's inequality,

$$
\begin{align}
\|M_t\|_{x_t'}^* &= \sqrt{M_t^\top \nabla^2(\mathcal{R}(x_t'))^{-1} M_t} \tag{24} \\
&= \sqrt{\mathbb{E}_t[d\sigma_t(y_t)\mathbf{A}_t^{-1}\mu_t]^\top \nabla^2(\mathcal{R}(x_t'))^{-1}\mathbb{E}_t[d\sigma_t(y_t)\mathbf{A}_t^{-1}\mu_t]} \tag{25} \\
&= \sqrt{d^2\mathbb{E}_t[\sigma_t(y_t)\mu_t]^\top \mathbf{A}_t^{-1}\mathbf{A}_t^2\mathbf{A}_t^{-1}\mathbb{E}_t[\sigma_t(y_t)\mu_t]} \tag{26} \\
&= \sqrt{d^2\mathbb{E}_t[\sigma_t(y_t)\mu_t]^\top \mathbb{E}_t[\sigma_t(y_t)\mu_t]} \tag{27} \\
&\leq \sqrt{d^2\mathbb{E}_t[\sigma_t^2(y_t)\mu_t^\top \mu_t]} \tag{28} \\
&\leq \sqrt{d^2\epsilon^2} \tag{29} \\
&= d\epsilon. \tag{30}
\end{align}
$$

Then we bound $\|x_\delta^{*'} - x_t'\|_{x_t'}$. From the triangle inequality,

$$
\|x_\delta^{*'} - x_t'\|_{x_t'} \leq \|x_\delta^{*'}\|_{x_t'} + \|x_t'\|_{x_t'}.
$$

By Lemma 1 and Lemma 4, we obtain $\|x_\delta^{*'}\|_{x_t'} \leq 2\nu$, $\|x_t'\|_{x_t'} = \sqrt{\nu}$ .

So $\|x_\delta^{*'} - x_t'\|_{x_t'} \leq 2\nu + \sqrt{\nu}$ and $\sum_{t=1}^T M_t^\top (x_\delta^{*'} - x_t') \leq Td\epsilon(2\nu + \sqrt{\nu})$. Then bound $\sum_{t=1}^T \mathbb{E}_t[g_t]^T(x_t' - x_\delta^{*'})$
By Lemma 7,

$$
\begin{align}
\sum_{t=1}^T \mathbb{E}_t[g_t]^T(x_t' - x_\delta^{*'}) &= \mathbb{E}_t\{\sum_{t=1}^T g_t^\top(x_t' - x_\delta^{*'})\} \\
&\leq \mathbb{E}_t\{\sum_{t=1}^T [g_t^\top x_t' - g_t^\top x_{t+1}'] + \frac{1}{\eta}[\mathcal{R}(x_\delta^{*'}) - \mathcal{R}(x_1')]\} \\
&\leq \mathbb{E}_t\{\sum_{t=1}^T [\|g_t\|_{x_t'}^* \|x_t' - x_{t+1}'\|_{x_t'}]\} + \frac{1}{\eta}(\mathcal{R}(x_\delta^{*'}) - \mathcal{R}(x_1')).
\end{align}
$$

Lemma 5 implies that $\|x_t' - x_{t+1}'\|_{x_t'} \leq 4d\eta$ is true by choice of $\eta$. Additionally, from Eq. (17), we deduce that $\|g_t\|_{x_t'}^* \leq d$. Therefore,

$$
\|g_t\|_{x_t'}^* \|x_t' - x_{t+1}'\|_{x_t'} \leq 4\eta d^2, \tag{31}
$$

$$
\mathbb{E}_t\{\sum_{t=1}^T [\|g_t\|_{x_t'}^* \|x_t' - x_{t+1}'\|_{x_t'}]\} \leq 4\eta d^2 T. \tag{32}
$$

With Lemma 3,

$$
\frac{1}{\eta}(\mathcal{R}(x_\delta^{*'}) - \mathcal{R}(x_1')) \leq \frac{\nu \log(\frac{1}{\delta})}{\eta}. \tag{33}
$$

Combine everything, we get

$$
\sum_{t=1}^T \theta_t^\top x_t - \sum_{t=1}^T \theta_t^\top x_\delta^* \leq 4\eta d^2 T + \frac{\nu \log(\frac{1}{\delta})}{\eta} + Td\epsilon(2\nu + \sqrt{\nu}). \tag{34}
$$

$\square$

Now we are ready to prove Theorem 1.

### 5.3 Proof of Theorem 1

*Proof.* Recall $\epsilon$-approximately linear function can be write as: $f(x) = \theta^\top x + \sigma(x)$. Thus, the regret of SCRIBLE with lifting algorithm

$$
\begin{aligned}
\mathbb{E}[\sum_{t=1}^T f_t(y_t) - \sum_{t=1}^T f_t(x^*)] &= \mathbb{E}[\sum_{t=1}^T [\theta_t^\top y_t + \sigma_t(y_t)] - \sum_{t=1}^T [\theta_t^\top x^* + \sigma_t(x^*)]] \\
&= \mathbb{E}[\sum_{t=1}^T \theta_t^\top y_t - \sum_{t=1}^T \theta_t^\top x^*] + \mathbb{E}[\sum_{t=1}^T \sigma_t(y_t) - \sum_{t=1}^T \sigma_t(x^*)].
\end{aligned}
$$

Firstly, we bound the front of the above equation,

$$
\mathbb{E}[\sum_{t=1}^T \theta_t^\top y_t - \sum_{t=1}^T \theta_t^\top x^*] = \sum_{t=1}^T \mathbb{E}[\theta_t^\top y_t] - \sum_{t=1}^T \mathbb{E}[\theta_t^\top x_t] + \sum_{t=1}^T \mathbb{E}[\theta_t^\top x_t] - \sum_{t=1}^T \mathbb{E}[\theta_t^\top x^*].
$$

From the Law of total expectation, we know

$$
\begin{aligned}
\sum_{t=1}^T \mathbb{E}[\theta_t^\top y_t] - \sum_{t=1}^T \mathbb{E}[\theta_t^\top x_t] &= \sum_{t=1}^T \mathbb{E}[\theta_t^\top (y_t - x_t)] \\
&= \sum_{t=1}^T \mathbb{E}[\mathbb{E}_t[\theta_t^\top (y_t - x_t)]] \\
&= \sum_{t=1}^T \mathbb{E}[\mathbb{E}_t[\theta_t^\top (\mathbf{A}_t \mu_t)]] \\
&= \sum_{t=1}^T \mathbb{E}[\theta_t^\top \mathbb{E}_t[(\mathbf{A}_t \mu_t)]] \\
&= \sum_{t=1}^T \mathbb{E}[\theta_t^\top \mathbf{0}] \\
&= \mathbf{0}.
\end{aligned}
$$

Thus,

$$
\mathbb{E}[\sum_{t=1}^T \theta_t^\top y_t - \sum_{t=1}^T \theta_t^\top x^*] = \mathbb{E}[\sum_{t=1}^T \theta_t^\top x_t - \sum_{t=1}^T \theta_t^\top x^*]. \tag{35}
$$

From Lemma 8, we have

$$
\begin{aligned}
\mathbb{E}[\sum_{t=1}^T \theta_t^\top x_t - \sum_{t=1}^T \theta_t^\top x^*] &\leq \mathbb{E}[4\eta d^2 T + \frac{\nu \log(\frac{1}{\delta})}{\eta} + Td\epsilon(2\nu + \sqrt{\nu}) + \delta DGT] \\
&\leq 4\eta d^2 T + \frac{\nu \log(\frac{1}{\delta})}{\eta} + Td\epsilon(2\nu + \sqrt{\nu}) + \delta DGT.
\end{aligned}
$$

Since $\sigma_t$ is chosen after knowing the player's action, it can cause as large a perturbation as possible. We using $|\sigma_t(x)| \leq \epsilon$ to bound $\sum_{t=1}^T \mathbb{E}[\sigma_t(y_t) - \sum_{t=1}^T \sigma_t(x^*)] \leq 2T\epsilon$ and combination of everything, we get

$$
\begin{aligned}
Regret &= \mathbb{E}[\sum_{t=1}^T f_t(y_t) - \sum_{t=1}^T f_t(x^*)] \\
&\leq 4d\sqrt{2\nu T \log T} + \frac{GD}{T} + Td\epsilon(2\nu + \sqrt{\nu}) + 2T\epsilon,
\end{aligned}
$$

where $\eta = \frac{\sqrt{2\nu \log T}}{2d\sqrt{T}}, \delta = \frac{1}{T^2}$. $\square$

### 5.4 Proof of Theorem 2

To establish the high-probability regret bound, we first introduce the necessary Lemma 9.

**Lemma 9** (Theorem 2.2. in Lee et al. (2020)). *Let $X_1, ..., X_T$ be a martingale difference sequence with respect to a filtration $F_1 \subseteq ... \subseteq F_T$ such that $\mathbb{E}[X_t \mid F_t] = 0$. Suppose $B_t \in [1, b]$ for a fixed constant $b$ is $F_t$-measurable and such that $X_t \leq B_t$ holds almost surely. Then with probability at least $1 - \gamma$ we have*

$$\sum_{t=1}^{T} X_t \leq C(\sqrt{8V ln(C/\gamma)} + 2B^* ln(C/\gamma)), \tag{36}$$

*where $V = \max\{1; \sum_{t=1}^{T} \mathbb{E}[X_t^2 \mid F_t]\}$, $B^* = \max_{t \in [T]} B_t$, and $C = \lceil logb \rceil \lceil log(b^2 T) \rceil$.*

The analysis in Lee et al. (2020) employs Lemma 9 to derive a high-probability bound for $\sum_{t=1}^{T}[y_t^\top \theta_t - x_t^\top g_t + u^\top (g_t - \theta_t)]$. In contrast, our approach defines $X_t = \theta_t^\top y_t - \theta_t^\top x_t$ and derives the high-probability bound for $\sum_{t=1}^{T}(\theta_t^\top y_t - \theta_t^\top x_t)$. This distinction in the application of Lemma 9 enables us to derive a tighter high-probability upper bound for bandit linear optimization.

With the support of Lemma 8 and Lemma 9, we are ready to prove Theorem 2.

*Proof.* Let $X_t = \theta_t^\top y_t - \theta_t^\top x_t$, then $\mathbb{E}_t[X_t] = \mathbb{E}_t[\theta_t^\top y_t - \theta_t^\top x_t] = 0$, $X_t = \theta_t^\top y_t - \theta_t^\top x_t \leq \|\theta_t\|\|y_t - x_t\| \leq GD$ and

$$
\begin{aligned}
\mathbb{E}_t[X_t^2] &= \mathbb{E}_t[(\theta_t^\top y_t - \theta_t^\top x_t)^2] \\
&= \mathbb{E}_t[(\theta_t^\top y_t)^2 + (\theta_t^\top x_t)^2 - 2\theta_t^\top y_t \theta_t^\top x_t] \\
&= \mathbb{E}_t[(\theta_t^\top y_t)^2] + \mathbb{E}_t[(\theta_t^\top x_t)^2] - \mathbb{E}_t[2\theta_t^\top y_t \theta_t^\top x_t] \\
&= \mathbb{E}_t[(\theta_t^\top y_t)^2] - \theta_t^\top x_t \theta_t^\top x_t \\
&\leq (1 + \epsilon)^2.
\end{aligned}
$$

Then,

$$
\begin{aligned}
\sum_{t=1}^{T} f_t(y_t) - \sum_{t=1}^{T} f_t(x^*) &= \sum_{t=1}^{T}[\theta_t^\top y_t + \sigma_t(y_t)] - \sum_{t=1}^{T}[\theta_t^\top x^* + \sigma_t(x^*)] \\
&\leq \sum_{t=1}^{T} \theta_t^\top y_t - \sum_{t=1}^{T} \theta_t^\top x^* + \sum_{t=1}^{T} \sigma_t(y_t) - \sum_{t=1}^{T} \sigma_t(x^*) \\
&\leq \sum_{t=1}^{T} \theta_t^\top y_t - \sum_{t=1}^{T} \theta_t^\top x^* + 2T\epsilon \\
&= \sum_{t=1}^{T} \theta_t^\top y_t - \sum_{t=1}^{T} \theta_t^\top x_t + \sum_{t=1}^{T} \theta_t^\top x_t - \sum_{t=1}^{T} \theta_t^\top x^* + 2T\epsilon.
\end{aligned}
$$

From Lemma 8, we know

$$\sum_{t=1}^{T} \theta_t^\top x_t - \sum_{t=1}^{T} \theta_t^\top x^* \leq 4d\sqrt{2\nu T \ln T} + \frac{GD}{T} + Td\epsilon(2\nu + \sqrt{\nu}), \tag{37}$$

where $\eta = \frac{\sqrt{2\nu \ln T}}{2d\sqrt{T}}, \delta = \frac{1}{T^2}$. Then by Lemma 9,

$$\mathbb{P}(\sum_{t=1}^{T}(\theta_t^\top y_t - \theta_t^\top x_t) \leq C(\sqrt{8V \ln(C/\gamma)} + 2B^* \ln(C/\gamma))) \geq 1 - \gamma, \tag{38}$$

where $V = (1 + \epsilon)^2 T$, $B^* = b = GD$, and $C = \lceil \ln GD \rceil \lceil \ln((GD)^2 T) \rceil$. Combine everything to conclude the proof. $\square$

Add more
discussion for
this
section(Unorgani
zed ..
Theorems by
Reviewer TGdS)

Highlight why
this problem is
interesting
(Weakness by
Reviewer 6Zex
)

## 5.5 Application to black-box optimization

From online to offline transformation, the result of this paper can also apply to black-box optimization for $\epsilon$-approximately linear function. This problem is important in that previous theoretical analyses for black box optimization can only deal with linear/convex/smooth objectives in the adversarial environments (via bandit convex optimization). So, it is quite meaningful to clarify the possibility of the black box optimization problems without such restrictions. In fact, our objective is not linear, nor smooth, even with a simple assumption.

Let $\hat{x}$ be the output of algorithm 1, then from Theorem 1. we can easily prove and ensure $f(\hat{x}) - \min_{x \in \mathcal{K}} f(x) \leq \frac{4d\sqrt{2\nu \ln T}}{\sqrt{T}} + \frac{GD}{T^2} + d\epsilon(2\nu + \sqrt{\nu}) + 2\epsilon$. Additionally, we provide a lower bound $2\epsilon$ for this problem (see Lemma 10). We can see that the difference between the lower bound and the upper bound is only $d\epsilon(2\nu + \sqrt{\nu})$ as $T$ approaches infinity. This suggests the potential existence of "easier" settings between the adversarial environment and the standard stochastic environment, where better algorithms might be found. It also motivates us to explore these settings further.

## 6 Lower bound

In this section, we show a lower bound of the regret. To do so, we consider a black-box optimization problem for the set $\mathcal{F}$ of $\epsilon$-approximately linear functions $f : \mathcal{K} \to \mathbb{R}$. In the problem, we are given access to the oracle $O_f$ for some $f \in \mathcal{F}$, which returns the value $f(x)$ given an input $x \in \mathcal{K}$. The goal is to find a point $\hat{x} \in \mathcal{K}$ such that $f(\hat{x}) - \min_{x \in \mathcal{K}} f(x)$ is small enough. Then, the following statement holds.

**Lemma 10.** *For any algorithm $\mathcal{A}$ for the black-box optimization problem for $\mathcal{F}$, there exists an $\epsilon$-approximately linear function $f \in \mathcal{F}$ such that the output $\hat{x}$ of $\mathcal{A}$ satisfies*

$$f(\hat{x}) - \min_{x \in \mathcal{K}} f(x) \geq 2\epsilon. \tag{39}$$

*Proof.* Firstly, suppose that the algorithm $\mathcal{A}$ is deterministic. At iteration $t = 1, ..., T$, for any feedback $y_1, ..., y_{t-1} \in \mathbb{R}$, $\mathcal{A}$ should choose the next query point $x_t$ based on the data observed so far. That is,

$$x_t = \mathcal{A}((x_1, y_1), ..., (x_{t-1}, y_{t-1})). \tag{40}$$

Assume that the final output $\hat{x}$ is returned after $T$ queries to the oracle $O_f$. In particular, we fix the $T$ feedbacks $y_1 = y_2 = \cdots = y_T = \epsilon$. Let $z \in \mathcal{K}$ be such that $z \notin \{x_1, ..., x_T, \hat{x}\}$. Then we define a function $f : \mathcal{K} \to \mathbb{R}$ is as

$$f(x) = \begin{cases} \epsilon, & x \neq z, \\ -\epsilon, & x = z. \end{cases} \tag{41}$$

The function $f$ is indeed an $\epsilon$-approximately linear function, as $f(x) = 0^\top x + \sigma(x)$, where $\sigma(x) = \epsilon$ for $x \neq z$ and $\sigma(x) = -\epsilon$ for $x = z$. Further, we have

$$f(\hat{x}) - \min_{x \in \mathcal{K}} f(x) \geq 2\epsilon. \tag{42}$$

Secondly, if algorithm $\mathcal{A}$ is randomized. It means each $x_t$ is chosen randomly. We assume the same feedbacks $y_1 = y_2 = \cdots = y_T = \epsilon$. Let $X = \{x_1, ..., x_T, \hat{x}\}$. Then, there exists a point $z \in \mathcal{K}$ such that $P_X(z \in X) = 0$, since $\mathbb{E}_{z'}[P_X(z' \in X | z')] = P_{z',X}(z' \in X) = \mathbb{E}_X[P_{z'}(z' \in X | X)] = 0$, where the expectation on $z'$ is defined w.r.t. the uniform distribution over $\mathcal{K}$. For the objective function $f$ defined in (41), we have $f(\hat{x}) - \min_{x \in \mathcal{K}} f(x) \geq 2\epsilon$ while $f$ is $\epsilon$-approximately linear. $\square$

**Theorem 3.** *For any horizon $T \geq 1$ and any player, there exists an adversary such that the regret is at least $2\epsilon T$.*

*Proof.* We prove the statement by contradiction. Suppose that there exists a player whose regret is less than $2\epsilon T$. Then we can construct an algorithm for the blackbox optimization problem from it by feeding the

online algorithm with $T$ feedbacks of the blackbox optimization problem and by setting $\hat{x} = \min_{t \in [T]} f(x_t)$. Then,

$$f(\hat{x}) - \min_{x \in \mathcal{K}} f(x) \leq \frac{\sum_{t=1}^{T} f(x_t) - \sum_{t=1}^{T} \min_{x \in \mathcal{K}} f(x)}{T} < 2\epsilon,$$

which contradicts Lemma 10. $\qquad\square$

This lower bound indicates that $\Omega(\epsilon T)$ regret is inevitable for the bandit optimization problem for $\epsilon$-approximately linear functions. We conjecture that the lower bound can be tightened to $\Omega(d\epsilon T)$, but we leave it as an open problem.

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
