# OpenReview forum: "Adversarial bandit optimization for approximately linear functions"
_TMLR — Rejected by TMLR_

### Review · Reviewer_6Zex · 2024-11-30

**Summary Of Contributions:**

This paper studies a new adversarial linear bandits under a small but arbitrary perturbation on the expected linear reward. The authors provide regret bounds for this problem and also extend their results to the offline setting.

**Audience:**

Yes

**Claims And Evidence:**

Yes

**Requested Changes:**

Please see the above weakness. I feel the draft can be drastically improved in terms of writing.

**Strengths And Weaknesses:**

Strength:
1. I am not an expert in adversarial bandits, but this problem setting seems new to me.
2. Theoretical proof are provided in detail.

Weaknesses:
1. I feel more related works should be provided, and the related work section is way too short. By reading the Lemmas, I feel most of them in this work are from existing literature such as Lee et al. Then the authors should clearly describe those line of work and highlight your main contributions over them.
2. Instead of jut putting the proof details, I feel authors should put more intuitions/descriptions for each theoretical results to connect them. The current version is more like an Appendix but not the main paper.
3. Has this problem been studied in the stochastic linear bandit? If so you should also list them in your related works, and argue the main technical challenges over them.
4. Some experimental results should be used to validate the efficiency of your proposed method. Like comparing your method with the linear adversarial bandit one under perturbations to show the robustness of your method.
5. It is also important to highlight why this problem is interesting itself. I can hardly image any real-world examples that can be formulated by "almost" linear reward setting but not the linear reward setting.

---

> ### Author Response · Authors · 2025-01-10
> **Thank you very much for all your comments and questions. We will address all of them here.**
>
> About "1.I feel more related works should be provided, and the related work section is way too short. By reading the Lemmas, I feel most of them in this work are from existing literature such as Lee et al. Then the authors should clearly describe those line of work and highlight your main contributions over them":
>
> I will add explanations of SCRIBLE and SCRIBLE with lifting and increasing learning rates to help readers better understand the content referenced from other papers and how my work differs from previous research.
>
> About "Instead of just putting the proof details, I feel authors should put more intuitions/descriptions for each theoretical results to connect them. The current version is more like an Appendix but not the main paper.":
>
> I will reorganize Section 4 and add appropriate explanations for each lemma and theorem.
>
> About "Has this problem been studied in the stochastic linear bandit? If so you should also list them in your related works, and argue the main technical challenges over them.":
>
> This problem has not been studied before. However, in the stochastic linear bandit, there is a similar setup where the loss function is a linear function plus an error term that follows a specific distribution (usually with an expectation of zero). We do not include more references because this paper does not adopt methods from the stochastic linear bandit.
>
> About "Some experimental results should be used to validate the efficiency of your proposed method. Like comparing your method with the linear adversarial bandit one under perturbations to show the robustness of your method.":
>
> Our main contribution lies in addressing a new problem using a novel analytical approach to derive theoretical results. The algorithm presented in the paper is merely a simplification of existing algorithms. Therefore, we do not conduct experimental comparisons.
>
> About "It is also important to highlight why this problem is interesting itself. I can hardly image any real-world examples that can be formulated by "almost" linear reward setting but not the linear reward setting.":
>
> This problem is important in that previous theoretical analyses for black box optimization can only deal with linear/convex/smooth objectives in the adversarial environments (via bandit convex optimization). So, it is quite meaningful to clarify the possibility of the black box optimization problems without such restrictions.  In fact, our objective is not linear, nor smooth, even with a simple assumption

---

### Review · Reviewer_TGdS · 2024-12-01

**Summary Of Contributions:**

This work studies the adversarial bandit for approximately linear functions. Such function class consists of functions that can be seen as linear functions with adversarial, bounded perturbations. The work provides an algorithm with in-expectation and high probability regrets as well as a lower bound for the problem.

**Audience:**

Yes

**Broader Impact Concerns:**

No, it's a fundamental and theoretical work on optimization.

**Claims And Evidence:**

No

**Requested Changes:**

The work would benefit from further revisions to improve clarity in several areas. Please see the previous section.

**Strengths And Weaknesses:**

## Strengths

The work can potentially provide some inspiration to the field in its problem formulation and methods.

## Weaknesses

The work would benefit from further revisions to improve clarity in several areas before discussing its contributions.

### Literature Review
The literature review could be more comprehensive. In the introduction, the literature is presented somewhat arbitrarily, with discussions jumping between topics. At times, the focus is more on the cost in T, and suddenly more to the dimension d in relation to the function class, and sometimes on high-probability results. For instance, it is unclear how or why the discussion transitions from the T^{3/4} bound by Flaxman et al. (2005) to an information-theoretic bound in 2020, and then moves to a specific algorithm, SCRIBLE, with high-probability guarantees. Additionally, it is unclear why there appears to be a different set of references in the Introduction and the Related Works sections. Also, if you are adapting Lee et al. (2020), you should give a more detailed background along this line of research.

Just by the writing itself, it is not clear how you would place yourself in the literature, both in terms of problem formulation and adaptation of methods.

### Clarity in Methods

From a high-level perspective, it is unclear why some of the methods you mention are unsuitable and why your proposed method is expected to perform better. In Section 4, you state that method A is impossible and method B is not helpful, leading to the adoption of method C, but the reasoning is not well-developed or clearly articulated. Similar instances of vague explanations appear throughout the text and would benefit from further clarification and revision.

Additionally, the presentation does not seem sufficiently self-contained, as readers may need to refer to the papers and algorithms you cite to know what and how much you adapt from other papers. It would be helpful to provide more context or explanations within the paper itself to ensure the discussion is accessible to the audience.

### Unorganized Lemmas and Theorems

I highly recommend revising Section 4 to improve the organization and clarity of the presented results. The statements of lemmas and theorems are not well-organized and appear to be placed somewhat arbitrarily. Please consider creating subsections to group related results and ideas for better readability and logical flow.

Please add necessary explanations between the statements to clarify the logic of why each result is introduced and how it connects to the overall argument. For instance, you could create a subsection to discuss the nice properties of the \nu-normal barrier, which play an important role in your results.

Several results were taken directly from previous works. To improve the presentation, you could, for example, state that you are about to present your result A, which requires result B from prior works. This would be the appropriate moment to introduce and formally state result B, which would make the introduction of the result more purposeful.

Additionally, some results are based on or adapted from previous works. For instance, Lemma 6 doesn’t require the full proof to be re-expanded. It would be clearer to state the original result and explain how you adapted it for your specific case. The same strategy could likely be applied to Lemma 5.

Since Lemma 7 is only used to prove Theorem 2, I suggest creating a subsection for these together, including all necessary discussions to keep the logic self-contained. Additionally, Lemma 9 is the key to get your theorems. Provide a more high-level discussion on how it comes, and explain how it contributes to your main results.

In the proof of Lemma 9, references like “the proof of Lemma 3 implies ...” are unclear, as Lemma 3 contains many steps. Specify exactly which part of Lemma 3 you are referring to.

Adding corollaries to Theorems 1 and 2 would help readers better understand their potential implications. For instance, you could state: “If condition A holds, the regret is X; if condition B holds, the regret is Y.” This would clarify how different terms contribute to the results and under what conditions one term dominates over others.

Finally, Section 4.1 is too brief. I see that you want it to bridge between the upper bounds and the lower bound you are presenting, but it is not well-explained.

### Lower Bound

The lower bound seems to imply that for an \epsilon-approximately linear function class, there is nothing we can do to prevent suffering a linear dependence on \epsilon. I wonder how this result and problem setting suggests and indicates in the field.

---

> ### Author Response · Authors · 2025-01-10
> **Thank you very much for all your comments and questions. We will address all of them here.**
>
> About literature Review:
>
> I will rewrite the literature review and add a subsection in Chapter 2 to describe the work of Lee et al. (2020) and the differences from our work.
>
> About Clarity in Methods:
>
> I will add explanations of SCRIBLE and SCRIBLE with lifting and increasing learning rates to help readers better understand the content referenced from other papers and how my work differs from previous research. I will revise the reasoning behind why 'method A is impossible, method B is not helpful, and thus method C is adopted.'
>
> About unorganized Lemmas and Theorems:
>
> I will reformat Section 4 and dedicate a subsection to discussing the $\nu$-normal barrier. Regarding the lemmas adapted from previous results, I will remove the proofs and add the original results, as per your suggestion, explaining how they were adapted from the original work.
>
> I will create a subsection specifically for the proof of Lemma 7 and Theorem 2. Additionally, I will move Theorems 1 and 2 to the beginning of Section 4, as they represent the main contributions of the paper.
> I will clarify the references in Lemma 9 by directly pointing to the specific equations from Lemma 3 being cited.
>
> About "Adding corollaries to Theorems 1 and 2 would help readers better understand their potential implications. For instance, you could state: “If condition A holds, the regret is X; if condition B holds, the regret is Y.” This would clarify how different terms contribute to the results and under what conditions one term dominates over others.":
>
> The conditions required to derive Theorems 1 and 2 are exactly the same. Therefore, we do not add any additional corollaries.
>
> About "Finally, Section 4.1 is too brief. I see that you want it to bridge between the upper bounds and the lower bound you are presenting, but it is not well-explained.":
>
> I will add more explanations regarding this part.
>
> ABout "Lower Bound":
> The lower bound clarifies the limits of algorithms for the $\epsilon$-approximately linear function class. Thus the lower bound is useful for ensuring the optimality of algorithms in our adversarial setting. Also,  our lower bound motivates the exploration of "easier" settings between our adversarial setting and the standard stochastic settings where better algorithms exist.

---

> > ### Comment · Reviewer_TGdS · 2025-02-03
> >
> > I appreciate the authors' efforts in revising the draft according to the feedback. The readability has improved, but there is still room for further enhancement. At this point, it’s honestly challenging to pinpoint specific areas for improvement. As suggested by Reviewer iDsE, I recommend an overall reconsideration of the paper’s structure.
> >
> > Additionally, while it’s helpful that you provide arguments for why certain Lemmas are needed and reference related quantities, these quantities sometimes appear several pages earlier, making it hard for readers to recall. You might consider numbering the lines where these quantities are introduced and removing line numbers where they are not referenced—this would help readers navigate the content more easily.

---

### Review · Reviewer_iDsE · 2024-12-26

**Summary Of Contributions:**

The paper studies an online bandit optimization problem over $T$ rounds, where the player selects an action $x_t$ from a bounded, closed convex action space $\mathcal{ K} \subseteq \mathbb{R}^d $, and the adversary chooses a loss function $ f_t $. The player then incurs a loss $ f_t(x_t)$.
The paper assumes that the class of loss functions is  $\epsilon$-approximately linear, defined as $ f(x) = \theta_f^\top x + \sigma_f(x) $, where $\|\theta_f\| \leq G$ for some $G$ and $|\sigma_f(x)| \leq \epsilon$ for all  $ x \in \mathcal{K}$.

**Contributions**
For the case $ϵ \neq 0$, the paper proposes a modification of the SCRIBLE with lifting (Lee et al., 2020) and utilizes the properties of the $ν$-normal barrier (Nesterov & Nemirovskii, 1994; Nemirovski, 2004) to prove its high probability regret bound $O(\sqrt{T}+\epsilon T)$, and they also obtain its expected regret $O(\sqrt{T \ln T}+\epsilon T)$. For the case of no perturbation, they also propose improved regret bound over previous Lee et al. (2020)’s results. For the lower bound, they proved $\Omega(\epsilon T)$.

**Audience:**

Yes

**Broader Impact Concerns:**

This paper is a theoretical work.

**Claims And Evidence:**

No

**Requested Changes:**

Overall, the paper has contributions but the writing quality of the paper has some issues, particularly in explaining the key ideas of the algorithm for handling non-linear functions and why it improves over Lee et al. (2020) for the case of $\epsilon=0$.


  The presentation lacks sufficient details on critical procedures and theoretical analyses that are central to the algorithm’s design and regret improvements.

**Strengths And Weaknesses:**

**Strength:**

The first high-probability regret bound for investigating  $\epsilon$-approximately linear function, as well as the lower bound analysis.

Improved regret bound for bandit linear optimization for the oblivious adversary.


**Weakness**


The following comments are not major weaknesses but improvement for writing the paper would be appreciated.

A. Writing Quality

1. Description of Algorithm 1: The main text relies heavily on pseudocode and lacks a general explanation of SCRIBLE.
For instance, the rationale behind sampling the exploration direction $\mu_t$ from the lifting space, and the reason of choice of the $\nu$-self-concordant barrier $R$ and the definition of the loss estimator $g_t$ remain unexplained in the main text (Section 4) and only put in Algorithm 1.
The readers are supposed to know the details of SCRIBLE algorithm (Abernethy et al., 2008), but the authors should explain at least key procedures and ideas of SCRIBLE. Also, Section 1 is only used to mention the previous regret bound and related setting; explanations for techniques from (Abernethy et al., 2008) and further development by (Lee et al., 2020) to achieve high probability and data-dependent regret bound are completely missing in the main text.


2. Description of main theorem: Why don’t you describe Theorem 1 first rather than presenting other Lemmas 1~8?

3. Other technical presentation: $\sigma_t$ and $\theta_t$ have not been formally defined in Section 3.2 and they directly appear in the analysis presented in Section 4.


**B: Missing some references**

It would be helpful to position the proposed algorithms more clearly in the context of existing literature, especially SCRIBLE and its variants, bandit linear optimization, high-probability regret bounds etc. For instance, some references could be cited in the revised version.

Jacob D. Abernethy, Elad Hazan, and Alexander Rakhlin. 2012. Interior-Point Methods for Full-Information and Bandit Online Learning. IEEE Trans. Inf. Theor. 58, 7 (July 2012), 4164–4175

Peter L Bartlett, Varsha Dani, Thomas Hayes, Sham Kakade, Alexander Rakhlin, and Ambuj Tewari. High-probability regret bounds for bandit online linear optimization. In Conference On Learning Theory, 2008.

Gergely Neu. 2015. Explore no more: improved high-probability regret bounds for non-stochastic bandits. In Proceedings of the 28th International Conference on Neural Information Processing Systems - Volume 2 (NIPS'15). 3168–3176.

**C: Some Questions for Comparison with [Lee et al.  2020]**

In the algorithm by Lee et al. (2020), the learning rate is increased $ O(d \log (\nu T)) $ times to control the eigenvalues of the matrix $ A_t $, which in turn helps to reduce the variance of the unbiased estimators. On the other hand, the proposed method claims to avoid increasing the learning rate, leading to an improvement in the regret bound by $ O(d \log (\nu T)) $. Is this understanding correct?

The reviewer finds it unclear why such a variance control becomes unnecessary in the proposed method. Although Lemma 5 explains the property of $ g_t $ as an unbiased estimator, this appears to be a critical aspect of both the algorithm design and the theoretical analysis. Also, Lemma 5 is a novel result for handling non-linear functions. Yet, Section 4, where the algorithm is introduced, does not adequately explain its benefit. If this is one of the major differences from Lee et al. (2020), it should be highlighted earlier in the paper. It also appears that this difference enables the removal of the need for increasing the learning rate, leading to the improved regret bound when $ \epsilon = 0 $.

Lastly, was there a need to increase the learning rate in Lee et al. (2020) due to handling adaptive adversaries? If so, is it true that the improvement in your method is possible because it only handles oblivious adversaries and cannot be extended to adaptive adversaries?

**Minor typo:**
p.3 and Lemma 5/6/9: algorithm 1 →Algorithm 1

p.3:  int(\mathcal{K}’)

p.4:. z is not defined

Lemma 5: the sentence is not grammatically correct.

---

> ### Author Response · Authors · 2025-01-10
> **Thank you very much for all your comments and questions. We will address all of them here.**
>
> About A. Writing Quality:
>
> I will add a detailed description of the steps involved in the SCRIBLE algorithm and the SCRIBLE with lifting and increasing learning rates. I will also explain why their approach cannot be directly applied to our problem and how our analysis method improves the high-probability regret bound.
>
> I will restructure the paper to present Theorem 1 and Theorem 2 first before presenting Lemmas.
> I will add a more detailed problem setup in Section 3 ,including $\theta_{t}$ and $\sigma_{t}$.
>
> About B: Missing some references:
>
> Thank you very much for your recommendation. I will consider adding these references in Section 2.
>
> About C: Some Questions for Comparison with [Lee et al. 2020]
>
> Fistly, I answer your question about "In the algorithm by Lee et al. (2020), the learning rate is increased
> ...Is this understanding correct?"
>
> No, the main reason my analysis leads to an improvement in the regret bound is that I adopted a different regret analysis method compared to Lee et al. The learning rate is not the fundamental reason for this change. For the analysis of the regret bound of linear function, Lee et al. divide the regret mainly into two parts: $\sum_{t=1}^{T}[y_{t}^{\top}\theta_{t}-x_{t}^{\top}g_{t}+u^{\top}(g_{t}-\theta_{t})]$ and $\sum_{t=1}^{T}(y_{t}-u)^{\top}g_{t}$, where $u\in\mathcal{K}$.
> They provides a bound for $\sum_{t=1}^{T}(y_{t}-u)^{\top}g_{t}$ and a high-probability bound for $\sum_{t=1}^{T}[y_{t}^{\top}\theta_{t}-x_{t}^{\top}g_{t}+u^{\top}(g_{t}-\theta_{t})]$, which helps derive high-probability regret bounds for both oblivious and adaptive adversaries.
> We divide the regret into parts $\sum_{t=1}^{T}(y_{t}-x_{t})^{\top}\theta_{t}$  and $\sum_{t=1}^{T}(x_{t}-u)^{\top}\theta_{t}$ rather than following the approach of Lee et al. we provide a bound for $\sum_{t=1}^{T}(x_{t}-u)^{\top}\theta_{t}$ and and obtain a high-probability bound for $\sum_{t=1}^{T}(y_{t}-x_{t})^{\top}\theta_{t}$. This means that,, for an oblivious adversary, calculating the regret does not require considering the variance of the estimator $g_{t}$ and $\theta_{t}$, but only the variance between $y_{t}$ and $x_{t}$. This difference is a key factor that enables us to achieve a better high-probability regret bound.
>
> For the analysis of the regret bound of approximately linear function, we give a $\lVert h \lVert_{\nabla^2\mathcal{R}(x'_{t})}\leq2\nu $, where $ h\in\mathcal{K'}$ (Lemma 4),  instead of using the properties of the $ \nu $-normal barrier to control
>
> $\lVert h \lVert_{\nabla^2\mathcal{R}(x'_{t})}\leq$
>
> $ -\lVert h\lVert_{\nabla^2\mathcal{R}(x'_{t+1})}+\nu\ln (\nu T+1) $
>
> as done in Lee et al.'s paper. This result is key to bound the regret of the approximately linear function.
> In fact, in Lee et al.'s paper, the increase in the learning rate was solely aimed at controlling $\lVert h\lVert_{\nabla^2\mathcal{R}(x'_{t})}$. That is why we no longer need to adjust the learning rate.
>
>
>
>
>
> The proof of Lemma 5 is based on the analysis of SCRIBLE, Lee et al.'s paper(Lemma B.14) also provides a similar proof. the core idea is to bound  $\lVert x_{t}^{'}-x_{t+1}^{'}\ \lVert_{\nabla^2\mathcal{R}(x'_{t})}$ in order to bound
>
> $\lVert x_{t}^{'}-h \lVert_{\nabla^2\mathcal{R}(x'_{t})}$.
>
> Regarding whether 'the need to increase the learning rate in Lee et al. (2020) was due to handling adaptive adversaries?'—Yes, increasing the learning rate helps in handling adaptive adversaries. Our improved analysis is currently applicable only to the oblivious adversary setting.
>
> I will add more explanation about the paper [Lee et al. 2020] and provide a more detailed comparison between the methods used in their paper and my own.
>
> About "Minor typo:": I will revise these problems.

---

> ### Comment · Reviewer_iDsE · 2025-01-20
>
> Thank you very much for your detailed responses and the revised version of the manuscript. I appreciate the effort you have put into addressing the comments and making significant improvements.
>
> I believe that the simplified regret analysis compared to Lee et al. (2020), as well as the new approach considering the influence of $\sigma(x_t)$, are well-articulated and provide a clear contribution. To further enhance the manuscript, I recommend adding a high-level overview of these crucial ideas to the introduction. This would help readers quickly grasp the core contributions of your work before reading detailed technical discussions.
>
> Additionally, while the revised version is much improved, there are still minor typos. I encourage you to carefully proofread the manuscript to ensure it is free of such issues.

---

> ### Author Response · Authors · 2025-01-24
> **Thank you very much for all your comments and advices. We will address all of them here.**
>
> I'm glad that the improvements to the paper made it easier for you to read.
>
> We have submitted a new revised version, adding a high-level overview in the introduction. We also corrected some minor typos in the paper.
>
> If you have any questions or suggestions, feel free to let us know. We will respond and make changes as quickly as possible.

---

> ### Comment · Reviewer_iDsE · 2025-01-27
>
> Thank you very much for your revisions. There are still some points to be improved.
>
>  1. In Section 3, the statement "...and the adversary simultaneously chooses a linear function $f_t(x_t) (=\theta_t ^{\top} x_t)$" might be misleading.  The adversary first chooses the loss vector $\theta_t$​ and then chooses the perturbation $\sigma_t$​. As a result, $f_t(x_t)=\theta_t^{\top} x_t +\sigma_t (x_t)$.  The sentence should be modified.
>
>
>  2. Section 4: I believe Section 4 is not clear enough, especially the explanation of proposed algorithms and main results. The structure or the order of the algorithm and the differences in the analysis could be improved. The algorithm could first be explained clearly and succinctly with the necessary notation, and then the differences from [Lee et al. 2020]  can be explained both in terms of the algorithm and the analysis. Additionally, in Section 4.3, the benefits and interpretations of Theorems 1 and 2 should be highlighted. Almost all of the reviewers asked for additional explanation of the analysis, which was likely intended to be written in the analysis (Section 5 in the current version), not Section 4. The authors can improve the structure of the paper.
> Also, when introducing detailed equations, it would be better to specify which lemmas or theorems from [Lee et al. 2020] or other work you are referring to.
>
> Overall:
> Despite many additions and improvements after the review comments, there are still issues with the writing. I recommend the authors re-consider the structure of the paper and submit it to a similar conference or journal, or consult with the AC to ensure that a proper final version is submitted.
>
> Typo:
> There is unnecessary space before the footnote in Introduction.
>
> Some points require a space before the citation: For example "e.g.Dani et al."

---

### Decision · Action_Editor_EMLy · 2025-03-02

**Recommendation:** Reject

**Comment:**

I have personally found the problem studied by the authors to be interesting, and the contribution to be quite reasonable. Some reviewers have shared the same sentiment as well. Despite the appreciation of some of the technical content, the initial round of reviews have criticized several other aspects of the paper, especially when it came to the quality of writing, and the motivation & contextualization of the problem setting. While the updated version provided by the authors has addressed some of these concerns, the reviewers have ultimately remained unconvinced about the suitability of this work for publication at TMLR.

Besides these issues, I have identified another serious problem with this work. In addition to the missing references pointed out by the reviewers, there appears to be some further significant gaps in the literature review. In particular, I expected to see some discussion of the very well-publicized negative results of Du et al (2019, https://arxiv.org/abs/1910.03016), and the follow-up discussion by Lattimore et al. (2020, https://arxiv.org/abs/1911.07676) and Dong and Van Roy (2019, https://arxiv.org/abs/1911.07910). The results in these works show that finding an near-optimal action in misspecified linear bandit problems can incur an exponential computational and statistical cost in terms of the problem dimensionality, which seems to be in contradiction with the results of the present paper. Indeed, it appears that the results of this work can be used in a standard online-to-batch conversion scheme to obtain a $2\varepsilon$-optimal action in time polynomial in $d$. The lower bounds in the works cited above suggest that this should not be possible for a constant better than $\sqrt{d}$ in general, which raises some serious concerns about the validity of the results claimed in this submission. Even if there is something wrong with my argument above, the paper cannot be published without providing a detailed discussion of the line of work mentioned above, ideally explaining why the lower bounds do not apply to the setting studied in the paper.

Given these problems, I have no option but to reject this paper. That said, I would like to encourage the authors to keep pursuing this line of work, and try to reconcile the mismatch between their results and the existing literature.

**Audience:**

The topic of the paper is relevant for a large community of learning theorists interested in bandit problems.

**Claims And Evidence:**

This is a theoretical paper, whose main result is a theorem that is presented along its complete proof. None of the reviewers raised concerns about the correctness of the proofs, but my own reading has revealed some potential issues (see below).

**Resubmission Of Major Revision:**

The authors may consider submitting a major revision at a later time.